# Toxicity of Zinc Oxide Nanoparticles on the Embryo of Javanese Medaka (*Oryzias javanicus* Bleeker, 1854): A Comparative Study

**DOI:** 10.3390/ani11082170

**Published:** 2021-07-22

**Authors:** Naweedullah Amin, Syaizwan Zahmir Zulkifli, Mohammad Noor Amal Azmai, Ahmad Ismail

**Affiliations:** 1Department of Biology, Faculty of Science, Universiti Putra Malaysia, UPM Serdang, Selangor 43400, Malaysia; sodes.amin123@gmail.com (N.A.); mnamal@upm.edu.my (M.N.A.A.); aismail@upm.edu.my (A.I.); 2Department of Zoology, Faculty of Biology, Kabul University, Kart-e-Char, Kabul 1006, Afghanistan; 3International Institute of Aquaculture and Aquatic Sciences (I-AQUAS), Universiti Putra Malaysia, Batu 7, Jalan Kemang 6, Teluk Kemang, Port Dickson 71050, Malaysia; 4Environment & Social Performance, Group Health, Safety Security and Environment, Petroliam Nasional Berhad (PETRONAS), Kuala Lumpur 50088, Malaysia; 5Aquatic Animal Health and Therapeutics Laboratory, Institute of Bioscience, Universiti Putra Malaysia, UPM Serdang, Selangor 43400, Malaysia

**Keywords:** nanoparticle, zinc oxide, embryo, nanotoxicity, Javanese medaka

## Abstract

**Simple Summary:**

In recent years, the production and distribution of ZnO NPs have gradually increased. As the number of ZnO NPs containing products grows, and the release of these products into the environment—particularly to the aquatic environment—has increased, several questions about their toxic effects on aquatic organisms have arisen. In this study, we explore the embryotoxicity of ZnO NPs by using the newly introduced model organism *Oryzias javanicus* (Javanese medaka). We found that the 96 h LC_50_ of ZnO NPs on the embryo of Javanese medaka were 0.643 mg/L, 1.333 mg/L, and 2.370 mg/L in ultra-pure, deionized, and dechlorinated tap water. The toxicity of ZnO NPs increased as both the concentration and time of exposure increased. The results of this study demonstrate that ZnO NPs are extremely toxic for the early life stage of Javanese medaka.

**Abstract:**

(1) Background: Zinc oxide nanoparticles (ZnO NPs) are widely applied in various human products. However, they can be extremely toxic for aquatic organisms, particularly fish. This research was conducted to determine the LC_50_ of ZnO NPs on the embryos of Javanese medaka (*Oryzias javanicus*) in ultra-pure, deionized, and dechlorinated tap water; (2) Methods: The experiments were conducted in a completely randomized design (CRD) with three replicates for six treatments for acute (0.100, 0.250, 0.500, 1.00, 5.00, and 10.00 mg/L) exposures for each type of water; (3) Results: The LC_50_ of ZnO NPs at 96 h was determined as 0.643 mg/L in ultra-pure water, 1.333 mg/L in deionized water, and 2.370 in dechlorinated tap water. In addition to concentration-dependent toxicity, we also observed time-dependent toxicity for ZnO NPs. In addition, the sizes of ZnO NPs increased immediately after dispersion and were 1079 nm, 3209 nm, and 3652 nm in ultra-pure, deionized, and dechlorinated tap water. The highest concentration of measured Zn^2+^ in exposure concentrations was found in ultra-pure water, followed by deionized and dechlorinated tap water suspensions. Furthermore, Javanese medaka showed high sensitivity to acute exposure of ZnO NPs in all types of water.

## 1. Introduction

Currently, several dangerous chemicals are considered a global threat to humans, other organisms and also the environment itself. Nevertheless, from time to time, the world is constantly generating and introducing large amounts of chemical substances into the environment. At the same time, the effects on organisms and the environment of these substances are not well known. Among evolving chemicals, nanoparticles (NPs) are one of those which are described as a particle with at least one dimension between 1 and 100 nm with different characteristics from bulk materials [1], and nanotechnology is known as the use of these materials [2]. Nanotechnology has recently developed as a rapidly growing market with efficient effects on major economic sectors with novel and unique properties that have been used in a diverse group of consumer goods such as agriculture, cosmetics, electronics, textiles, and pharmaceuticals [3,4,5]. Based on their composition, NPs can be classified into carbon-based NPs (carbon nanotubes and carbon black), inorganic NPs (generally this type of NPs contain metals (Al, Bi, Co, etc.) and metal oxides (ZnO, CuO Al_2_O_3_, etc.)), organic-based NPs (this type of NPs involve dendrimers, micelles, liposomes, and polymer NPs which are produced mainly through organic material, excluding carbon-based NPs and inorganic NPs), and composite-based NPs which are produced from the combination of NPs with other NPs or NPs with bulk materials (such as hybrid nanofibers) [5,6]. Among several NPs, ZnO NPs are known as one of the most efficiently used in the nano-scale range with a wide bandgap and large excitonic binding energy [7], high stability, anticorrosion and photo-catalytic properties [8], non-migratory, fluorescent, piezoelectric, absorptive, and scatters ultraviolet light [9], diverse nanostructures [10], and antimicrobial activity [11]. Zinc oxide NPs are already extensively implemented in consumer goods such as paints, UV filters, biosensors, paper, plastics, ceramics, building materials, rubber, power electronics, coatings, feed, photocatalytic, degradation of textiles, and printed matter [12,13]. ZnO NPs, which are the third most widely applied metal-based NPs with an approximate world-wide total production of 550 to 33,400 tons [5], can reach the environment, particularly the aquatic environment by (1) wastewater which contains the highest amount of ZnO NPs (0.3–0.4 μg/L), (2) direct use and (3) deposition from the air compartment [14,15].

Once ZnO NPs are released to the aquatic environment, changes in their physico-chemistry occur, which modifies their environmental fate and toxicity to aquatic species. These changes mostly lead to a decrease in bioavailability and toxicity, although increases in bioaccumulation and toxicity were reported in some cases. One of the most important impacts on the fate of MeO-NPs in an aquatic ecosystem is the formation of a cluster of NPs called an aggregation/agglomeration. These alterations caused changes in the wide range of size distributions of ZnO NPs by forming aggregates, and some studies have reported that aggregations change the size of ZnO NPs 10-fold bigger than the primary ZnO NPs size [5,8]. The formation of aggregates of ZnO NPs are correlated to various parameters such as the presence of dissolved organic matter (DOM) in the surrounding media [12], dispersion method [8], pH, ionic strength [9], physico-chemical properties of ZnO NPs including particle size, shape [16], and surface properties such as surfaces modification [17]. Aggregation of ZnO NPs in the aquatic ecosystem will facilitate settling from suspension onto the bottom surface of the water body. Due to increased aggregation and sedimentation, estuarine and aquatic sediments have been proposed to be the endpoint for several NPs [5]. Most of the sedimentation of ZnO NPs started once they reached the water and occurred within 24 h, and then the sedimentation processes decreased over time [18]. Furthermore, Poynton et al. [15] showed that 97% of ZnO NPs dispersed in water settled out, and about 2% of Zn^2+^ was dispersed in the water. Another important transformation of ZnO NPs is their dissolution in the aquatic environment, which strongly affects their behavior in the environment. Surprisingly, data on the solubility of ZnO NPs are limited, and because of different laboratory conditions, such as the suspension medium, pH, salinity, dispersion methods, DOM, ionic strength and particle surface area, the reports appeared contradictory [1,3,10,15].

Although ZnO NPs are widely produced and released in various industries, due to their toxic impacts on different aquatic organisms, they have been listed as extremely toxic [19]. Meanwhile, studies related to ZnO NPs toxicity on aquatic vertebrate organisms have concentrated largely on fish, in particular zebrafish. Several reports have shown that ZnO NPs can be highly toxic to zebrafish, particularly in the early developmental stages [20,21]. Furthermore, serious threats and higher toxicity compared to other NPs in aquatic environments have been reported in recent studies for ZnO NPs. For instance, Zhu et al. [22] reported that ZnO NPs showed higher toxicity compared to TiO_2_ NPs and Al_2_O_3_ NPs on the early life stage of zebrafish. Another study that has shown ZnO NPs are more toxic than TiO_2_ NPs is the study of Bhuvaneshwari et al. [23], who reported 27.62 and 71.63 mg/L for ZnO NPs and 117 and 120.9 mg/L for TiO_2_ NPs as 48 h LC_50_ on *Artemia salina* under pre-UV-A and visible light conditions. Meanwhile, the idea that water chemistry can affect the fate and behavior of chemicals, and their subsequent bioavailability to fish, is well established but, to date, there have been no systematic studies of the toxicity of ZnO NPs to the same species of organism in different types of water.

*Oryzias javanicus* (Javanese medaka) belongs to the Adrianichthyidae family [24]. This species is widely distributed in Asian countries and highly adaptable to fresh, brackish, and saltwater [25]. The sensitivity of the species belonging to this family makes it an ideal test organism for toxicology and ecotoxicology studies. Recent studies have indeed used Javanese medaka as the test organism because of their high adaptability to both freshwater and saltwater, broad geographical range and availability throughout the year [26,27,28], short life span and life cycle, fast development [29,30], hardy, easy to identify and cultivate, short spawning period <1 min, and their transparent eggs [31]. These properties make it a suitable choice for studies, especially studies that involve early life stages. Hence, this study was conducted to determine the median lethal concentration (LC_50_) of ZnO NPs in ultra-pure, deionized, and dechlorinated tap water on the embryo of Javanese medaka.

## 2. Materials and Methods

### 2.1. Source of ZnO NPs and Test Organisms

ZnO NPs (#677450 zinc oxide nanopowder, <50 nm particle size (BET), purity >97%) were purchased from Sigma-Aldrich, Missouri, United States. Adult Javanese medaka were collected from the estuary area in Sepang, Selangor, Malaysia. The fish were caught using scoop nets and immediately brought back to the laboratory and acclimatized in dechlorinated tap water with a 14 h/10 h light/dark cycle for at least 21 days. After acclimatization, the sexing process was carried out under a dissecting microscope. The sex of the fish was differentiated based on the characteristics described by Imai et al. [32]. Two males and four females were put in 3 L tanks and were maintained through the circulating system for breeding purposes. Consequently, to make them spawn daily, the photoperiod was maintained at 14 h light and 10 h dark, temperature (28–30 °C), dissolved oxygen (5.5–7.5 mg/L), pH (5.5–6.5), salinity (0 ppt). Fish were fed with newly hatched *Artemia nauplii* (Brine shrimp) larvae, since most recent studies reported that feeding Javanese medaka with newly hatched Brine shrimp resulted in active spawning with a high number of eggs [30,33,34].

### 2.2. Physicochemical Characterization of ZnO NPs

For size verification, X-ray diffraction (XRD) was performed. The shape of ZnO NPs was visualized by transmission electron microscopy (TEM JEM-2100F, JEOL Ltd., Tokyo, Japan). Samples for TEM analysis were prepared from 10 mg/L suspensions of ZnO NPs on carbon-coated copper grids. Measurement of zeta potential and size distribution was carried out on solutions of ZnO NPs (1 mg/L) prepared in ultra-pure, deionized, and dechlorinated tap water, with samples taken rapidly after preparation. The size distribution and zeta potential were measured by dynamic light scattering (DLS) using the Malvern Zetasizer Nano-ZS instrument (Malvern Panalytical, Worcestershire, UK).

### 2.3. Measured Exposure Concentrations and Zinc Ion Release

The stock solutions of ZnO NPs (100 mg/L) were prepared in ultra-pure (18.2 MΩ cm), deionized, and dechlorinated tap water and then diluted to the exposure concentrations. To obtain a homogeneous suspension, the stock solutions were stirred with a magnetic stirrer for 30 min prior to use. In order to stimulate an environmentally relevant situation, the stirring method was used instead of utilizing surfactants or sonication. Range finding tests were conducted to determine the range for the acute toxicity tests. The nominal exposure concentrations of ZnO NPs in this study were (0.100, 0.250, 0.500, 1.00, 5.00, and 10.00 mg/L) in ultra-pure, deionized, and dechlorinated tap water. The concentrations of ionic Zn in ZnO NPs exposure concentrations were measured by ICP-MS. Samples for ICP-MS were taken rapidly after preparation. The ion release was shown as the measured zinc concentration (mg/L) of the total nominal exposure concentration of zinc.

### 2.4. Toxicity Tests

Experiments were conducted according to the Organization for Economic Cooperation and Development testing guidelines (OCED 2013) [35]. Newly spawned Javanese medaka egg clusters were carefully collected from the female’s body by hand, clusters of eggs were then separated with forceps and washed three times with saltwater, and the embryos at the blastula stage < 3 hpf were selected for further procedures. To start the 96 h exposure, the fertilized embryos were immediately transferred to 6-well plates, each well containing 10 embryos and 10 mL of exposure solution. Each well was one replicate and each concentration contained three replicates. Static toxicity tests were conducted after 24 h embryos were checked under a stereomicroscope (Olympus CX31 2D, Tokyo, Japan) and dead embryos were removed and recorded. Meanwhile, exposure solutions were removed and new stock and diluted solutions were prepared freshly. The light/dark ratio was 14 h/10 h and was maintained during the exposure. The water parameters were recorded throughout the experiments as follows: temperature at 26 ± 1 °C, salinity at 0 ppt, pH at 6.90 ± 0.4, and dissolved oxygen at 5.53 ± 0.6 mg/L.

### 2.5. Statistical Analysis

To determine the LC_50_, probit analyses were performed using log concentration in GraphPad prism version 8.0.2 for Windows (GraphPad Software, La Jolla, CA, USA).

## 3. Results

### 3.1. Characterization of ZnO NPS

X-ray diffraction result confirmed the ZnO NPs with a crystal structure of 26 nm in primary size (Figure 1A). The TEM images (Figure 1B) revealed that most of the ZnO NPs in suspension had a hexagonal shape. The average hydrodynamic diameters of the ZnO NPs were 1079 nm, 3209 nm, and 3652 nm and the observed zeta potentials were −6.43 mV, 3.04, and 2.01 mV in ultra-pure, deionized, and dechlorinated tap water (Table 1).

### 3.2. Dissolution of ZnO NPs

With increasing nominal concentration of ZnO NPs, the measured Zn^2+^ concentration also increased; however, different dissolution rates were observed for ZnO NPs in ultra-pure, deionized, and dechlorinated tap water. For instance, higher concentrations of Zn^2+^ were observed for ZnO NPs in ultra-pure water, which were 0.355 ± 0.064, 0.911 ± 0.010, 1.794 ± 0.136, 2.809 ± 0.154, 21.805 ± 0.417 and 42.235 ± 2.878 mg/L in the nominal concentrations (0.100, 0250, 0.500, 1.00, 5.00, and 10.00 mg/L, respectively) of ZnO NPs. However, at the same nominal concentrations, the measured Zn^2+^ concentrations were 0.917 ± 0.062, 1.054 ± 0.046, 1.454 ± 0.283, 3.073 ± 0.086, 20.295 ± 5.409 and 30.862 ± 0.860 mg/L, respectively, in deionized water and 0.711 ± 0.028, 1.211 ± 0.069, 2.457 ± 0.063, 3.485 ± 0.120, 7.935 ± 0.049 and 15.391 ± 0.268 mg/L, respectively, in dechlorinated tap water (Figure 2).

### 3.3. Embryotoxicity of ZnO NPs

The mortality of Javanese medaka embryos increased as the concentration of ZnO NPs increased. For instance, mortality rates of Javanese medaka embryos exposed to ZnO NPs were: 3.33, 20.00 30.00, 83.33, 90.00 and 100.00%; 3.33, 6.67, 10.00, 33.33, 93.33 and 100.00%; and 6.67, 10.00, 13.33, 23.33, 83.33 and 96.67% in ultra-pure, deionized, and dechlorinated tap water at 0.100, 0.250, 0.500, 1.00, 5.00 and 10.00 mg/L of ZnO NPs, respectively. This indicates that ZnO NPs had a concentration-dependent toxicity on the embryos of Javanese medaka in all types of water.

The lowest mortality was observed in 0.100 mg/L of ZnO NPs and did not increase throughout the experiment. Mortalities of Javanese medaka embryos exposed to ZnO NPs were less than 15% in all treatment groups at 24 and 48 h post-exposure (hpe). However, mortality increased sharply for 0.250, 0.500, 1.00, 5.00, and 10.00 mg/L of ZnO NPs at 72 and 96 hpe, which indicates that as well as concentration-dependent toxicity, toxicity increased with time for certain concentrations as shown in Figure 3.

The lowest 96 h LC_50_ value determined by probit analysis for ZnO NPs on Javanese medaka embryos was in ultra-pure water (0.6438 mg/L) followed by deionized water (1.333 mg/L), and dechlorinated tap water (2.370 mg/L) (Figure 4).

## 4. Discussion

In the present study, the embryotoxicity of ZnO NPs on Javanese medaka was tested. Among several NPs, ZnO NPs are known as one of the most efficiently used in the nano-scale range with their unique characteristics ideally found in various applications in consumer goods such as cosmetics, textiles, rubber and electronic industries [9,10]. However, studies have shown that ZnO NPs are extremely toxic to fish, particularly at the early life stage, which causes several morphological deformities [19,36].

The measured hydrodynamic diameters for ZnO NPs demonstrates that the hydrodynamic diameter of ZnO NPs in all types of water crossed the nanometer-scale immediately after dispersing, indicating that Javanese medaka embryos during these studies were exposed to both aggregates and well dispersed ZnO NPs. A similar finding was reported by Cong et al. [19], that the hydrodynamic diameter of ZnO NPs crossed the nanometer scale upon dispersion in filtered artificial saltwater. Furthermore, the observed zeta potential for ZnO NPs suspensions were between +30 mV and −30 mV, thereby suggesting that ZnO NPs suspensions were not stable based on the DLVO hypothesis [37], which revealed that the sedimentation of ZnO NPs aggregates had been unavoidable throughout the experiments.

Differential dissolution of ZnO NPs in different types of water might be due to the presence of different amounts of ionic strength. High dissolution rates at low ionic strength were reported for ZnO NPs in previous studies. For instance, Li et al. [3] studied the effects of water chemistry on the dissolution of ZnO NPs and their toxicity on *Escherichia coli* and demonstrated that ZnO NPs had a concentration-dependent dissolution, and dissolution of ZnO NPs was reduced in the existence of ionic strength. A similar result was reported in the studies of Keller et al. [38] and Fairbairn et al. [39] who reported that the concentration of Zn^2+^ decreased as a result of increasing ionic strength in suspension. Zinc ions and ZnO NPs are two types of elemental zinc that can exist in water, soil, and organisms. In terms of water persistence, though, they have entirely different properties. Since Zn^2+^ is intrinsically persistent, it can be transformed into other compositions and can form zinc complexes with ions found in water, such as zinc chloro complexes or zinc hydroxide. Zinc oxide NPs, on the other hand, are not always persistent. Since the different types of water contain different types of ions, such as chloride, sulfate, and organic matter, an increase in ionic strength typically raises the degree of metal complication. Li et al. [9] compared the solubility of ZnO NPs in freshwater and saltwater and reported that ZnO NPs showed higher solubility in saltwater compared to freshwater, but due to an increase in ionic strength, the concentration of Zn^2+^ was lower in saltwater than in freshwater at the same concentration. Therefore, the solubility of ZnO NPs in different types of water was different.

Concentration-dependent toxicity effects of ZnO NPs were reported in several recent studies. For instance, Zhu et al. [22] showed that during the 96 h exposure, the hatching of zebrafish embryos was not affected by 0.5 mg/L of ZnO NPs; however, the toxicity increased as the concentration of ZnO NPs increased, indicating that the toxicity of ZnO NPs had a concentration-dependent property. Bai et al. [10] on the other hand, reported that 50 mg/L and 100 mg/L of ZnO NPs caused mortality, and 1–25 ZnO NPs affected the hatching rate of zebrafish embryos in the E3 medium which ultimately caused abnormality during the 96 h exposure. Time-dependent toxicity of ZnO NPs was reported by Li et al. [9] who showed that the mortality rate of *Mugilogobius chalae* (yellow stripe goby) embryos in all ZnO NPs treatments was <20% on days 0–4, but increased sharply on day 5, at which time they increased to ≤50% for the 25 mg/L and 50 mg/L ZnO NPs concentration. Time-dependent toxicity for ZnO NPs was also reported on zebrafish embryos [10].

The 96 h LC_50_ value of ZnO NPs on Javanese medaka embryos differed in different types of water. The lowest 96 h LC_50_ (0.6438 mg/L) value was observed in ultrapure water followed by deionized water (1.333 mg/L) and dechlorinated tap water (2.370 mg/L). This decrease in toxicity might possibly be due to an increase in aggregate size in the presence of high ionic strength for ZnO NPs. The DLS results showed that the size distribution of ZnO NPs in dechlorinated tap water was three-fold and ~one-fold higher compared to in ultra-pure water and deionized water. This demonstrates that the aggregate size of ZnO NPs increased with increasing ionic strength, which eventually decreased the toxicity of ZnO NPs. Many previous studies reported similar results for ZnO NPs. For instance, Young et al. [40] reported that due to the increase in the ionic strength of the suspension by increasing salinity, the size of aggregates of ZnO NPs increased and eventually decreased their toxicity to the marine diatom (*Thalassiosira pseudonana*). Another study that reported the size of aggregates of ZnO NPs increased due to higher ionic strength by increasing salinity of suspension is the study of Park et al. [41] who demonstrated that increasing salinity of the suspension affected the aggregation and dissolution of ZnO NPs and CuO NPs, and decreased their acute toxicity on *Tigriopus japonicus*. This correlation between the size of aggregates of ZnO NPs and ionic strength might be due to the attraction of van der Waals force, which takes place as a result of the combination of the electric double layers of particles, leading the particles to form larger aggregates [42]. The stability of the ZnO NPs suspension is determined by the combined effects of van der Waals attraction and electrostatic repulsion caused by an electric double layer of cations surrounding ZnO NPs in an aqueous medium. The electrical charge carried by ZnO NPs causes mutual electrostatic repulsion between nearest particles, and ZnO NPs can stay separate and stable in suspension if this electrical charge overcomes the van der Waals attraction force. However, a suspension with a high ionic strength induces compression of the electrical double layer, which decreases the electrostatic repulsion forces between the NPs and, as a result, encourages the NPs to aggregate and settle out of suspension [43].

Dissolution of ZnO NPs is another phenomenon that might affect its toxicity to the embryos of Javanese medaka in different types of water. The dissolution of metallic oxide NPs serves a vital role in the determination of their toxicity. For instance, Auffan et al. [44] demonstrated that releasing metal ions from NPs can partially lead to their toxicity. Similar findings were reported in the study of Bai et al. [10]. Although the toxicity of ZnO NPs is predominantly related to the free Zn^2+^ in the aqueous solution, an increase in the ionic strength of the suspension could decrease the concentration of free Zn^2+^ and thus lower their toxicity. Similar findings were reported by Li et al. [3] who revealed that the concentration of Zn^2+^ in a ZnO NPs suspension decreased in the existence of ionic strength, which later on reduced their toxicity on *Escherichia coli*. Furthermore, the author also mentioned that during adsorption of Zn^2+^ by cells, the presence of cations in water such as (H^+^, Ca^2+^, Mg^2+^, Na^+^, and K^+^) may compete and thus lower the toxicity of ZnO NPs. According to the study of Hogstrand et al. [45] Zn^2+^ absorbs through a Ca^2+^ uptake mechanism and/or a Zn-specific absorption mechanism, where Ca^2+^ will inhibit Zn^2+^ uptake competitively in rainbow trout. Furthermore, Califford and McGeer [46] also demonstrate that the presence of Ca^2+^ and Mg^2+^ reduced the toxicity of Zn^2+^ on *Daphnia pulex*. A decrease in the toxicity of Zn^2+^ to the algae (*Pseudokirchneriella subcapitata*) in the presence of Na^+^, Mg^2+^, and Ca^2+^ was also reported by Heijerick et al. [47].

We have observed lower LC_50_ values for the embryos of Javanese medaka compared to the LC_50_ values reported in previous studies for ZnO NPs. For instance, Zhu et al. [22] reported the value of 96 h LC_50_ as 1.793 mg/L after exposing the embryos of zebrafish to ZnO NPs in Milli-Q^®^ water. Another study, which evaluated the acute toxicity of ZnO NPs at the early life stage of yellow stripe goby, by Li et al. [9], reported 45.40 mg/L as the 96 h LC_50_ value of ZnO NPs, which is higher than the LC_50_ value of this study. Furthermore, Saddick et al. [48] also reported that ZnO NPs were toxic to Nile tilapia and Red belly tilapia in deionized water, and calculated 5.5 mg/L and 5.6 mg/L as 96 h LC_50_. Other species of fish with different 96 h LC_50_ values have been summarized in Table 2.

Similar to our results, different 96 h LC_50_ values were reported for the same species of fish in different types of medium in different studies. Zhu et al. [22] reported 1.793 mg/L as 96 h LC_50_ for ZnO NPs on the embryo of zebrafish in Milli-Q^®^ water; however, Xiong et al. [20] reported 4.92 mg/L as a 96 h LC_50_ of ZnO NPs on the embryo of zebrafish in distilled water. However, Du et al. [49] studied the acute toxicity of ZnO NPs on zebrafish embryos in zebrafish culturing medium (E3 medium) and reported 60 mg/L as a 96 h LC_50_. Another study that reported that ZnO NPs were toxic toward zebrafish in early life stages (96-hpf) in pure water is the study of Wehmas et al. [50] who reported 2.20 mg/L as a 24-h LC_50_. However, in the same study, zebrafish embryos (8-hpf) were also exposed to the same ZnO NPs for 24 h and reported +50 mg/L as a 24 h LC_50_. Therefore, they concluded that as well as a medium dependent risk category, distinct life stages of the same species showed different sensitivities to the same ZnO NPs.

## 5. Conclusions

For the first time, we explored the toxicity of ZnO NPs and found that they are extremely toxic to Javanese medaka embryos. The 96 h of median lethal concentration (LC_50_) of the acute exposure of ZnO NPs were 0.6438 mg/L, 1.333 mg/L, and 2.251 mg/L in ultra-pure, deionized, and dechlorinated tap water. The mortality rate of Javanese medaka embryos increased as the concentration of ZnO NPs increased in all types of water, and there was a strong correlation between time of exposure and mortality of embryos. The toxicity of ZnO NPs was influenced by different types of water by affecting their size distribution, dissolution, and concentration of Zn^2+^ on the embryos of Javanese medaka. The LC_50_ value is a valuable criterion for nanoecotoxicity; it is not a representative concentration of contaminants in aquatic environments, but it is essential for indicating the toxicity of certain pollutants. Rather than using existing test organisms, Javanese medaka was used in this study as a new model organism of nanoecotoxicological exposure so that we could have existing indigenous organisms that live in tropical regions. As Javanese medaka show a higher sensitivity to ZnO NPs compared to other well-known model organisms, particularly zebrafish, the findings of this study can support the creation of the Javanese medaka as a novel model organism for tropical areas in aquatic nanoecotoxicological studies. In comparison to previous research conducted in saltwater, we observed high sensitivity for ZnO NPs in freshwater, which raises the importance of the physicochemical parameters of water on the toxicity of ZnO NPs. This indicates the need for a comparative study on the toxicity of ZnO NPs in freshwater versus saltwater. Moreover, further investigation needs to been carried out on the transcriptomic effects of ZnO NPs on this species.

## Figures and Tables

**Figure 1 animals-11-02170-f001:**
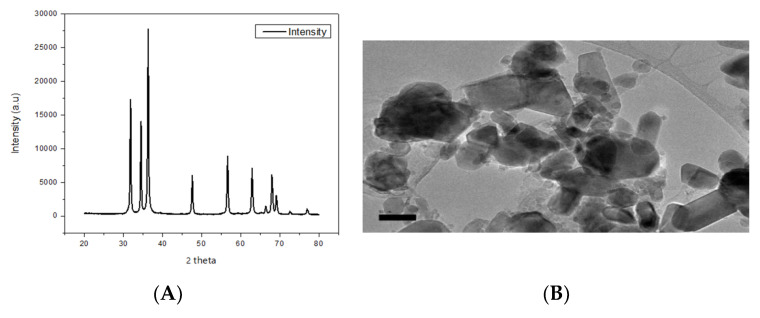
X-ray diffraction image (**A**) and transmission electron microscopy images (scale bar 50 nm) of ZnO NPs dispersed in ultra-pure water (**B**).

**Figure 2 animals-11-02170-f002:**
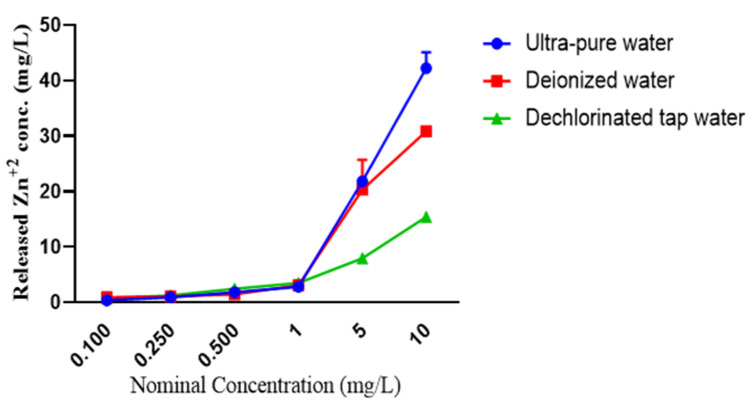
The concentrations of released Zn^2+^ from ZnO NPs in ultra-pure water, deionized water, and dechlorinated tap water.

**Figure 3 animals-11-02170-f003:**
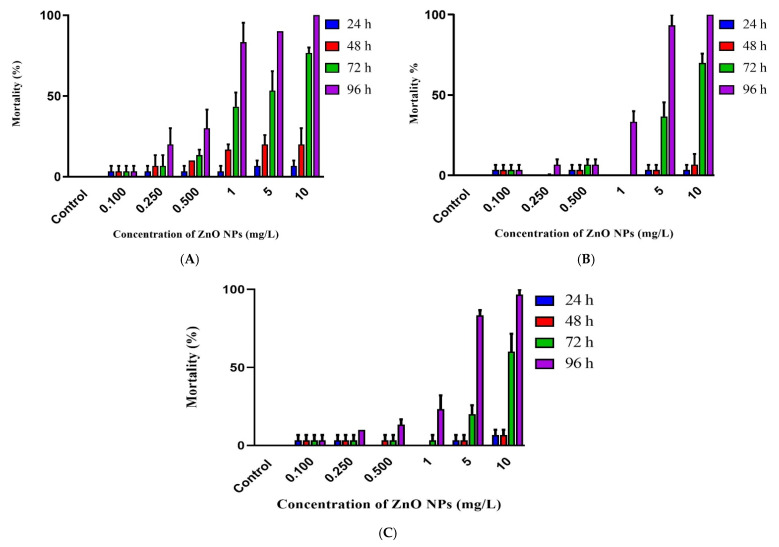
Mortality of Javanese medaka embryos at various time intervals following exposure to different concentrations of ZnO NPs in ultra-pure water (**A**), deionized water (**B**), and dechlorinated tap water (**C**).

**Figure 4 animals-11-02170-f004:**
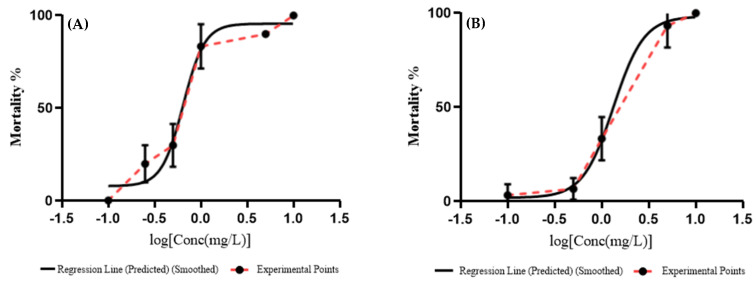
Graph of the mortality (%) against the concentration of ZnO NPs (mg/L) in (**A**) ultra-pure water, (**B**) deionized water, and (**C**) dechlorinated tap water. The lethal concentration (50%) value for ZnO NPs determined by probit analysis at different concentrations (0, 0.100, 0.250, 0.500, 1, 5, and 10 mg/L) for 96 h.

**Table 1 animals-11-02170-t001:** Measured zeta potential and size distribution of ZnO NPs in ultra-pure, deionized, and dechlorinated tap water.

Medium	Ultra-Pure Water	Deionized Water	Dechlorinated Tap Water
Zeta potential (mV)	−6.43	3.04	2.01
Size distribution (nm)	1079	3209	3652

**Table 2 animals-11-02170-t002:** Acute toxicity of ZnO NPs on different species of fish.

Species	Life Stage	Duration of Exposure (Hours)	Medium	Median Lethal Concentration (LC_50_) (mg/L)/(ppm)	Reference
*Oryzias Javanicus* (Javanese medaka)	Embryo	96	Ultra-pure water	0.6438	Present study
			Deionized water	1.333	
			Dechlorinated tap water	2.370	
*Danio rerio* (Zebrafish)	Embryo	96	Zebrafish culture medium	60	[49]
*Danio rerio* (Zebrafish)	Embryo	96	Drinking water	30.51	[51]
*Danio rerio* (Zebrafish)	Embryo	96	Distilled water	4.92	[20]
*Danio rerio* (Zebrafish)	Embryo	96	Milli-Q^®^ water	1.793	[22]
*Oncorhynchus mykiss* (Rainbow trout)	Adult	96	Dechlorinated tap water	25.50	[52]
*Oreochromis niloticu* (Nile tilapia)	Adult	96	Deionized water	5.5	[48]
*Coptodon zilli* (Red belly tilapia)				5.6	
*Labeo rohita*	Adult	96	Deionized water	31.15	[53]

## Data Availability

The corresponding author will provide the data used in this study upon request.

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
