# Peer review of "Toxicity of Zinc Oxide Nanoparticles on the Embryo of Javanese Medaka (*Oryzias javanicus* Bleeker, 1854): A Comparative Study"

_animals, 2021, doi:10.3390/ani11082170_

Round 1
Reviewer 1 Report
xxx
Author Response
Dear reviewer,
We would like to express our gratitude for the time you dedicated to reviewing our manuscript. Thank you very much for your time.

Reviewer 2 Report
Paper has improved a lot and authors have performed and important revision work. In my opinion, the new version is easy to read and interesting
Author Response

(The authors gave the same response as above.)

Reviewer 3 Report
Review comments on the manuscript by Amin et al
Toxicity of zinc oxide nanoparticles on the embryo of Javanese medaka (Oryzias javanicus Bleeker, 1854): a comparative study
The aim of the study was to assess the toxicity of zinc oxide nanoparticles using standard static 96 hour toxicity tests using embryos of a new test organism Javanese medaka. Toxicity was tested using three types of water to assess variability I toxicity between water types. They also undertook a physicochemical characterisation of the zinc oxide nanoparticles in the three water types.
Overall, the data presented achieves the study aims and provides an original contribution to our knowledge of the metals in fish. The analytical techniques and QA/QC procedures, sampling design and statistical analysis are appropriate to address the aims of the study and the tables and figures relevant to presenting the results of the study. As such, I consider that a revised paper incorporating the attached track change comments and addressing the comments detailed below could be resubmitted for assessment for publication.
There is no figure 4 in the manuscript. The text and figure captions should be revised to reflect this deleted figure.
Given that the author’s first language is not English, I have marked as track changes on the manuscript in the attached MS Word document, numerous suggested spelling and grammatical changes to improve the quality and flow of the manuscript. The authors should review these changes and have the manuscript reviewed before resubmission. The figures and tables will need to be reinserted in the attached revised manuscript.
There are also a number of issues with the references. The authors should check previous issues of the journal regarding presentation of references.

Author Response
Dear reviewer,
We would like to express our gratitude for the time you dedicated to reviewing our manuscript. Furthermore, we are grateful for the suggestions provided, which greatly improved the quality of the manuscript. They have been amended and corrected as suggested in the manuscript. Below you can find the responses to your comments.
Point 1: There is no figure 4 in the manuscript. The text and figure captions should be revised to reflect this deleted figure
Response 1: The figures and tables revised and reinserted. I would like to mention that our manuscript has 4 figures and 2 tables, there was a typo in the figure 4 caption in the previous version of the manuscript, which already corrected in the revised manuscript.
Point 2: Given that the author’s first language is not English, I have marked as track changes on the manuscript in the attached MS Word document, numerous suggested spelling and grammatical changes to improve the quality and flow of the manuscript. The authors should review these changes and have the manuscript reviewed before resubmission.
Response 2: The given track changes have been followed and changes were made in the revised manuscript accordingly.
Point 3: There are also a number of issues with the references. The authors should check previous issues of the journal regarding presentation of references.
Response 3: The references were checked and changes were made accordingly.
